# *FOXD1* Is a Transcription Factor Important for Uveal Melanocyte Development and Associated with High-Risk Uveal Melanoma

**DOI:** 10.3390/cancers14153668

**Published:** 2022-07-28

**Authors:** Quincy C. C. van den Bosch, Josephine Q. N. Nguyen, Tom Brands, Thierry P. P. van den Bosch, Robert M. Verdijk, Dion Paridaens, Nicole C. Naus, Annelies de Klein, Emine Kiliç, Erwin Brosens

**Affiliations:** 1Department of Ophthalmology, Erasmus MC Cancer Center, Erasmus MC University Medical Center Rotterdam, 3000 CA Rotterdam, The Netherlands; q.vandenbosch@erasmusmc.nl (Q.C.C.v.d.B.); j.nguyen@erasmusmc.nl (J.Q.N.N.); t.brands@erasmusmc.nl (T.B.); n.naus@erasmusmc.nl (N.C.N.); 2Department of Clinical Genetics, Erasmus MC Cancer Center, Erasmus MC University Medical Center Rotterdam, 3000 CA Rotterdam, The Netherlands; a.deklein@erasmusmc.nl; 3Department of Pathology, Section Ophthalmic Pathology, Erasmus MC Cancer Institute, Erasmus MC University Medical Center Rotterdam, 3000 CA Rotterdam, The Netherlands; t.vandenbosch@erasmusmc.nl (T.P.P.v.d.B.); r.verdijk@erasmusmc.nl (R.M.V.); 4Department of Pathology, Leiden University Medical Center, 2333 ZA Leiden, The Netherlands; 5The Rotterdam Eye Hospital, 3011 BH Rotterdam, The Netherlands; d.paridaens@oogziekenhuis.nl

**Keywords:** in silico analysis, transcription factor, melanocyte development

## Abstract

**Simple Summary:**

Despite successful treatment of primary uveal melanoma (UM), metastases still occur in approximately 50% of the patients. Unfortunately, little is known about the mechanism behind metastasized UM. By reanalyzing publicly available single-cell RNA sequencing data of embryonic zebrafish larvae and validating the results with UM data, we have identified five transcription regulators of interest: *ELL2*, *KDM5B*, *REXO4*, *RBFOX2* and *FOXD1*. The most significant finding is *FOXD1*, which is nearly exclusively expressed in high-risk UM and is associated with poor survival. *FOXD1* is a novel gene which could be involved in the metastatic capability of UM. Elucidating its function and role in metastatic UM could help to understand and develop treatment for UM.

**Abstract:**

Uveal melanoma (UM) is a deadly ocular malignancy, originating from uveal melanocytes. Although much is known regarding prognostication in UM, the exact mechanism of metastasis is mostly unknown. Metastatic tumor cells are known to express a more stem-like RNA profile which is seen often in cell-specific embryonic development to induce tumor progression. Here, we identified novel transcription regulators by reanalyzing publicly available single cell RNA sequencing experiments. We identified five transcription regulators of interest: *ELL2*, *KDM5B*, *REXO4*, *RBFOX2* and *FOXD1*. Our most significant finding is *FOXD1*, as this gene is nearly exclusively expressed in high-risk UM and its expression is associated with a poor prognosis. Even within the *BAP1*-mutated UM, the expression of *FOXD1* is correlated with poor survival. *FOXD1* is a novel factor which could potentially be involved in the metastatic capacity of high-risk UM. Elucidating the function of *FOXD1* in UM could provide insight into the malignant transformation of uveal melanocytes, especially in high-risk UM.

‡ These authors contributed equally to this work.

§ The Rotterdam Ocular Melanoma Study Group (ROMS) is a collaborative research group with members from the Rotterdam Eye Hospital, Departments of Ophthalmology, Pathology and Clinical Genetics, of the Erasmus MC, Rotterdam, the Netherlands.

## 1. Introduction

Ocular melanomas account for around 5% of all melanoma with uveal melanoma (UM) being the most prevalent. Arising from the uvea, UM shows a distinct mutation burden pattern when compared to malignant melanoma of the skin with a predominant absence of a UV mutational signature [1,2]. UM has an incidence of one to nine per million in Europe, with a lower incidence in the southern countries and a higher incidence in the northern countries [3]. Successful treatment aims to preserve the eye and vision by stereotactic radiotherapy, brachytherapy or proton therapy and enucleation in case of more advanced tumors [4]. Despite successful local tumor control, UM is a highly aggressive disease where in the end 50% of all patients develop metastases [3,5]. Whilst much is known about UM prognostication and driver mutations [6,7,8,9,10,11], mechanisms behind metastatic UM are largely unknown. Apart from the recently discovered therapy drug Tebentafusp [12], therapeutic options are still limited in patients with metastatic UM, and therefore it is crucial to investigate metastatic events in UM [13]. Tumor cells frequently take advantage of transcription factors to alter their gene expression [14], specifically during events such as epithelial-to-mesenchymal transition [15]. Which transcription factor is used often relies on the cell type the tumor originates from [14]. Despite the fact that all UMs arise from the uvea, high-risk UMs harbor an altered transcriptome compared to low-risk UMs [16,17]. In addition, others have shown that high-risk UMs also have an increased expression of transcription regulators [16,17,18]. Here, we hypothesized that high-risk UM utilizes developmental transcription regulators used in the early establishment of uveal melanocytes for its metastatic potential. However, even though there is a clear general understanding how melanocyte cell fate is programmed [19], little is known about the mechanism behind the establishment of uveal melanocytes and what separates them from cutaneous and mucosal melanocytes.

All melanocytes spread throughout the body follow a similar gene expression profile, yet animal studies suggest an independent mechanism for ocular pigmentation during development. For instance, knock-out (KO) mice models of Cell Division Cycle 42 (*Cdc42*), the Actin Related Protein 2/3 (*Arp2-3*) complex, Brahma-related gene 1 (*Brg1*), Myosin X (*Myo10*) and Rac Family Small GTPase 1 (*Rac1*) revealed white spotting of the skin as a result of migration or differentiation defects of cutaneous melanocytes [20,21,22,23,24]. However, pigmentation of the eyes remained normal in these models. This phenomenon is also seen in non-mammalian vertebrate models. Genetic disruption of microphthalmia-associated transcription factor A (*mitfa*) and roy orbison *(roy)* in zebrafish generated the *casper* zebrafish that completely lacks melanocytes and iridophores. Nevertheless, despite the genetic disruption of melanocytic core genes *casper* zebrafish harbor normally pigmented and functional eyes [25]. Ocular pigmentation in zebrafish development has been shown to depend on orthodenticle homolog (otx) transcription factors rather than on *mitfa* or microphthalmia-associated transcription factor B (*mitfb*) [26]. Furthermore, using RNA in situ *hybridization* (ISH) of pigmentation genes in zebrafish revealed genetic divergence after the teleost genome duplication. For example, the human genome Premelanosome protein (*PMEL*) is expressed in all melanocytes as it is necessary for pigment functionality [27], whereas zebrafish express premelanosome protein a (*pmela*) in all melanocytes but restrict the expression of premelanosome protein b (*pmelb*) to the eyes [28]. In order to gain insights in transcriptional factors involved in early uveal melanocyte development, we argue that the zebrafish is a useful model as there is available embryonic vertebrate developmental data and genes to specify melanocyte location (eye versus all melanocytes).

To identify uveal melanocyte specific transcription regulators we mined the zebrafish atlas [29], as this single-cell RNA-sequencing (scRNA-seq) dataset yields whole zebrafish larvae at different embryonic stages. We have extracted melanocyte clusters and isolated the transcription regulators that are unique to *pmelb* positive cells. After the zebrafish to human orthologue prediction, we inspected the expression of the identified embryonic transcription regulators in healthy human melanocytes and UM. With this approach we aimed to (1) identify transcription regulators involved in early uveal melanocyte development and to (2) assess if these genes are active in high-risk UM.

## 2. Materials and Methods

### 2.1. (sc)RNA-Seq Datasets and Identification of Melanocyte-Specific Markers Utilizing Public Datasets

scRNA-seq datasets of whole zebrafish [29] and the human eye (GSE135922) [30] were downloaded from the gene expression omnibus (GEO) database (https://www.ncbi.nlm.nih.gov/gds/, accessed on 24 April 2021) and used for gene expression analysis (Figure 1). The Cell Ranger count matrices acquired through the GEO Accession viewer were processed into count tables using the Seurat package [31] in R.

Analysis of public Single Cell datasets was performed using the Qiagen CLC Genomics Workbench (version 21.0.5), CLC Single Cell Analysis Module (Version 21.1) and Seurat (version 4.0). To import the data, the Cellranger Matrix exports (MEX) were used where possible, either directly or through conversion of other formats to MEX using Seurat. Where there was no compatible count data available, the reads were imported using the Workbench’s SRA download function. The Single Cell plugin comes with default pipelines for both count data and reads. The read workflow has three additional steps: annotating the reads with Cell and UMI information, trimming them and mapping them against a reference genome. Then, both pipelines continue with a quality control step before creating a normalized matrix of all samples in a dataset. All datasets were processed using the default settings and the genomes available in the CLC Genomics Workbench (Homo Sapiens Hg19 Ensembl version 75 & Danio Rerio GRCz11). The normalized matrices were then clustered based on 5000 highly variable genes, principal component analysis and Pearson’s correlation for measuring distance. Cluster data was grouped by the Leiden algorithm (resolution 0.1) and visualized as a UMAP. Additionally, clusters were sub-setted and exported for further analysis. Gene expression thresholding was determined by overall gene count distribution to include a multimodal binomial distribution of gene counts while excluding low abundant genes. This threshold was set to be at least 100 gene counts (Appendix A). The DRSC integrative ortholog prediction tool (DIOPT; Version 8.5) was used to identify human orthologues [32]. Human orthologues with high ranking were included for further analysis. Human orthologue genes were uploaded into the Ingenuity Pathway Analysis (IPA) software (Qiagen Bioinformatics, Spring release April 2022, QIAGEN Inc., https://digitalinsights.qiagen.com/IPA, accessed on 27 May 2022) [33]. IPA-curated knowledge was used to identify gene function to eventually perform a Core Analysis using default parameters, to predict the effects on pathways and biological functions.

### 2.2. Bulk RNA-Sequencing Dataset Analysis of the TCGA and ROMS Uveal Melanoma Cohort

The UM RNA-seq data of the 80 UM samples in the TCGA database was obtained from the Genomic Data Common (GDC) Portal (https://portal.gdc.cancer.gov/, accessed on 18 June 2021). We used whole-transcriptome sequencing data from 26 UM from the ROMS-cohort, which has been described previously by Smit et al. [17]. Our ethics committee and current informed consent does not allow sharing of individual patient or control genotype information in the public domain. Upon reasonable request, data access can be granted by the data access committee of the Department of Clinical Genetics of the Erasmus MC Rotterdam, the Netherlands. The mRNA library was sequenced on the Ion Proton sequencer (ThermoFisher Scientific, Waltham, MA, USA). Reads were trimmed using Cutadapt Version 3.4 [34], aligned to the human reference genome hg19 using HISAT2 version 2.1.0 [35] and aligned reads were counted using htseq-count Version 0.9.1 [36] and normalized using DESeq2. Trimmed mean per million (TMM) values were used to normalize sequencing depth across samples. For each gene, counts per million (CPM) were calculated. Whole-transcriptome samples (ROMS and TCGA) were included in the analysis when mutational status and survival data were available. Identified transcription regulators were then inspected in both cohorts with regards to their mutational status (EIF1AX-, SF3B1- or BAP1-mutation).

### 2.3. Immunohistochemistry (IHC) of FOXD1

Immunohistochemistry was performed with an automated, validated, and accredited staining system (Ventana Benchmark ULTRA, Ventana Medical Systems, Tucson, AZ, USA) using an ultraview Universal Alkaline Phosphatase Red Detection Kit (Ventana reference no. 760-501). Following deparaffinization and heat-induced antigen retrieval (CC1 for 32 min), the tissue samples were incubated for 32 min with the FOXD1 antibody (1:800 dilution, Abcam, Waltham, MA, USA, #129324). Counterstain was done by hematoxylin II stain for 12 min and a blue coloring reagent for 8 min according to the manufacturer’s instructions (Ventana Benchmark ULTRA, Ventana Medical Systems, Tucson, AZ, USA). For each slide, healthy kidney tissue was used as a positive control.

### 2.4. Statistical Analysis

Statistical analysis was performed using GraphPad Prism 9.0 (GraphPad Software, Inc., San Diego, CA, USA). Differences between mutational subtype were assessed using a one-way analysis of variance (ANOVA) with the Kruskal–Wallis test and Dunn’s multiple comparisons test. The survival analysis was generated using the Kaplan–Meier method and use of the Log-Rank test to find differences between the mutational subtypes. A *p*-value of <0.05 was considered statistically significant.

## 3. Results

### 3.1. Identification of Ocular Melanocytic Clusters in Whole Zebrafish Single-Cell RNA Sequencing

Zebrafish melanocytes were identified based on established core melanocytic markers: *mitfa*, dopachrome tautomerase (*dct*), tyrosinase (*tyr*), tyrosinase-related protein 1b (*tyrp1b*) and *pmela*. We analyzed the zebrafish atlas per developmental stage (24 h post fertilization (hpf), 48 hpf and 120 hpf, Figure 2a–c). Across three uniform manifold approximation and projections (UMAPs) we identified a total of seven melanocytic clusters that express the core factors with a sufficient amount of cells (>100 cells). Next, we separated ocular melanocytes from other melanocytes based on *pmelb* expression which revealed four *pmelb*+ clusters and three *pmelb*- clusters. To gain insight into ocular melanocyte development we extracted the read-counts per identified cluster (*pmelb*+ cluster in red, *pmelb*- clusters in blue, Figure 1e–g). In order to investigate the expression of a gene, we determined our threshold to have at least a total sum of 100 read counts based on the overall gene-count distribution. With this threshold we included a multimodal binomial distribution of gene counts while filtering out low-abundance genes (Appendix A). After setting our threshold, we extracted all genes expressed per melanocytic cluster and combined the different developmental stages to end up with a *pmelb*- and a *pmelb*+ gene list (Appendix A). We compared both clusters together, where core melanocytic genes are expressed in both clusters and *pmelb* is discriminatory between the clusters (Figure 3). A total of 279 genes were unique to *pmelb*- cells, 1817 genes are expressed in both clusters and 637 genes were unique to *pmelb*+ cells.

### 3.2. Orthologue Prediction, Function Annotation and Expression Validation in Human Melanocytes and UM

To validate 637 uniquely expressed genes in *pmelb*+ zebrafish cells in human melanocytes, we performed the DIOPT orthologue prediction [32]. A total of 478 genes were predicted to their human orthologues with high ranking scores (Appendix A). Next, we annotated gene function and identified associated pathways with this gene list using IPA analysis (Appendix A). We identified 49 genes involved in transcriptional regulation (10.2%), 107 enzymes (22.0%), 233 genes classified as ‘other’ (56.8%), and the remaining genes were spread out in multiple functions (Figure 4a). Pathway analysis shows association with cellular development, embryonic development and cancer. Next, we inspected the expression of all transcription regulators in healthy human melanocytes (GSE135922) [30] using the same scRNA-seq pipeline and thresholding as done before. In total we identified the expression of 21 transcription regulators in healthy melanocytes, illustrating that these genes are expressed in ocular melanocytes (Figure 4b). In addition, the expression of the 49 transcription regulators was analyzed in RNA-seq data from the The Cancer Genome Atlas (TCGA) and our Rotterdam Ocular Melanoma Study (ROMS) cohort (Figure 4c). We averaged the gene expression in CPM by combining all samples and found that also in UM the majority of transcription regulators are expressed. However, to gain insights into factors used in high-risk UM, we looked into the expression pattern per secondary driver mutation (BRCA1 associated protein 1 (*BAP1*), Splicing factor 3b Subunit 1 (*SF3B1*) and Eukaryotic Translation Initiation Factor 1A X-Linked (*EIF1AX*)).

### 3.3. RNA Expression Analysis of Transcription Regulator Genes of Interest in UM

We assessed the expression levels of transcription regulator genes of interest in UM in our own and publicly available UM patient cohorts (*n* = 26; ROMS-cohort, *n* = 56; TCGA cohort) of which the secondary driver gene status and survival was known [17]. After normalization and alignment, the CPM values of all genes were extracted and compared between UM subtypes. Gene expression was grouped into driver mutations (*EIF1AX*, n = 16; *SF3B1*, n = 27; *BAP1*, n = 39). Out of 49 transcription regulators we identified five genes associated with high-risk UM and poor prognosis. A negative correlation was found in Elongation Factor For RNA Polymerase II 2 (*ELL2*), Lysine Demethylase 5B (*KDM5B*) and REX4 Homolog, 3′-5′ Exonuclease (*REXO4*), where low expression is associated with poor prognosis (Figure 5 and Appendix A). Positive correlations were found by increased expression of RNA Binding Fox-1 Homolog 2 (*RBFOX2*) and Forkhead Box D1 (*FOXD1*) within the *BAP1*-mutated UM, which are associated with poor prognosis (Figure 5). Interestingly, *FOXD1* is significantly expressed within the *BAP1*-mutated UM compared to *SF3B1*-mutated or *EIF1AX*-mutated UM (Figure 5c). Next, we assessed if expression of *FOXD1* is correlated with overall survival of UM patients. In both cohorts, expression of *FOXD1* (CPM ≥ 1) is significantly correlated with a poor prognosis. The median survival of *FOXD1*-positive UM is 26.5 months when compared to 129.6 months in the *FOXD1*-negative UM group (Figure 5d, *p* = <0.0001). Due to the striking expression of *FOXD1* specifically in the *BAP1*-mutated UM, we investigated if overall survival based on *FOXD1* is due to BAP1 co-segregation (n = 39). Surprisingly, even within the *BAP1*-mutated UM, the expression of *FOXD1* is significantly correlated with a poor prognosis (Figure 5d, *p* = 0.0413). *BAP1*-mutated UMs with *FOXD1* expression have a median survival of 26.1 months when compared to 41.7 months in *BAP1*-mutated *FOXD1* negative UM. To validate our findings on protein levels, we next assessed FOXD1 levels on UM with immunohistochemistry.

### 3.4. FOXD1, ELL2, RBFOX2, KDM5B and REXO4 as and RNA-Based Biomarkers in Uveal Melanoma

As proof of principle that our identified genes are able to distinguish low-risk from high-risk UM, we extracted the 15 genes used in the gene expression profiling (GEP) test [37] and plotted them together with *FOXD1*, *ELL2*, *RBFOX2*, *KDM5B* and *REXO4.* The ROMS and TCGA cohort were sorted based on disomy 3 or monosomy 3 status, which, in line with published data, showed a clear low-risk UM (class 1, disomy 3) and high-risk UM (class 2, monosomy 3) based on the GEP test. Upregulation of *FOXD1* and *RBFOX2* clusters was similar to the commercially used upregulated genes *CDH1*, *ECM1*, *HTR2B* and *RAB31* in class 2 UM. The GEP test also consists of downregulated genes in class 2 UM, where *ELL2*, *KDM5B* and *REXO4* also cluster in a similar fashion (Figure 6).

### 3.5. Immunohistochemistry (IHC) of FOXD1 in Uveal Melanoma

For IHC analysis, we analyzed 30 UM samples of the ROMS cohort with confirmed mutational status (*EIF1AX*, n = 8; *SF3B1*, n = 9; *BAP1*, n = 9, *EIF1AX*+*SF3B1*, n = 2; wild-type (WT), n = 2) and matched RNA expression (discovery cohort). We were unable to stain the full RNA-seq cohort due to lack of tissue for RNA–protein confirmation of the same patient. All samples were stained with FOXD1 antibody (Figure 7a,b) and samples were scored as either positive or negative. The expression of FOXD1 on protein level proved to be nearly correlated with survival (Appendix A, *p* = 0.1472). To strengthen this analysis, we have added 29 additional UM samples (replication cohort) with confirmed mutational status (*EIF1AX*, n = 9; *BAP1*, n = 20). The replication cohort alone also showed a near correlation between FOXD1 expression and survival (Appendix A, *p* = 0.2340). However, when we combined both cohorts (discovery and replication cohorts, n = 59), a significant correlation between FOXD1 expression on protein level and poorer survival was assessed (Figure 7c, *p* = 0.0439). As seen in other studies [38], we found FOXD1 expression to be focused mainly in the cytoplasm.

## 4. Discussion

In this study, we identified multiple transcription regulators which are associated with early uveal melanocyte development and high-risk UM. We have determined uveal melanocytic clusters with a unique expression pattern when compared to cutaneous melanocytes. Expression of *pmelb* was unique to this melanocytic cluster when compared to other melanocytic clusters. To support our hypothesis that this cluster contains uveal melanocytes, the RNA ISH of *pmelb* elucidated spatial expression restricted to the eye, unlike *pmela* [28]. Due to the teleost duplication zebrafish underwent, the diversification of *pmelb* can be an interesting target to study ocular pigmentation without targeting cutaneous and/or mucosal melanocytes in zebrafish.

Downregulated expression of transcription regulators in high-risk UM could help identify potential tumor-suppressor genes, however only the downregulation of *ELL2* is associated with prostate cancer and multiple myeloma based on our current knowledge [39,40,41,42]. *KDM4B* is a well-studied lysine-specific histone demethylase that is known to be overexpressed in multiple cancers [43,44,45,46]. Our findings are discordant with the literature, where a lower expression is associated with a poor prognosis (Appendix A). Our finding could be due to *BAP1* co-segregation, creating a bias in the prognosis prediction. Unlike *KDM5B*, *REXO4* is much less well-investigated in cancer biology. Current literature illustrates overexpression in hepatocellular carcinoma that is associated with tumor progression and immune infiltration [47,48]. Further investigations are needed to unravel the role of *KDM5B* and *REXO4* loss in high-risk UM.

We identified the upregulation of two transcription regulators, *RBFOX2* and *FOXD1*, in high-risk UM. *RBFOX2* is known to be involved in cerebellar and heart development [49,50], and *FOXD1* is involved in retina development [51,52], but to our knowledge neither have been associated with melanocyte development prior to this study. *RBFOX2* is a well-known regulator of epithelial-to-mesenchymal transition [53], where the alternate splicing of pre-mRNA alters signal transduction via a wide range of genes depending on cellular context [54,55,56]. Further investigation of *RBFOX2* overexpression in UM could reveal novel targets for drug intervention of high-risk UM.

Our most significant finding, *FOXD1*, was identified in 2–5 day-old zebrafish larvae; yet this was not found in healthy human uveal melanocytes (Figure 4b). We hypothesized that these transcription factors are lost after the establishment of uveal melanocytes. The human dataset consisted of uveal melanocytes of healthy adults (age range: 54–92 years) which doesn’t reflect embryonic development, whereas the zebrafish dataset (2–5 days post fertilization) showed the embryonic stage of vertebrate development. The differences in developmental stages could be a reason as to why *FOXD1* was not expressed in the human dataset. scRNA-seq data of human fetal uveal melanocytes could confirm this hypothesis; however, these data are not available to our knowledge. We reanalyzed the data of IPSC-derived melanocytes during neural crest cell generation. *FOXD1* was not expressed in these in vitro models, mimicking early melanocyte development (data not shown). In addition, we inspected the expression of *FOXD1* in early fetal development and *FOXD1* is only expressed in skeletal muscle, retinal progenitors, Müller glia, photoreceptor and ganglion cells [57]. In this stage, there are no melanocyte cells present in the eye. Additionally, we inspected the recently published ocular dataset using the interactive single-cell portal (https://singlecell.broadinstitute.org/single_cell; accessed on 29 June 2022) [58]. In adulthood, a few ocular melanocytes show expression of *FOXD1*, whereas the RPE and fibroblasts show a strong expression of *FOXD1* (Appendix A). Aggressive tumors are known to have a higher degree of dedifferentiation, which has been correlated to reactivate expression of developmental transcription factors [59,60,61]. Therefore, we assessed the association between *FOXD1* as a transcription regulator and its potential role in UM. Analyzing two independent datasets, we determined *FOXD1* to be associated with poor prognosis in UM based on RNA expression. Interestingly, we identified *FOXD1* as a biomarker that divides *BAP1*-mutated UM into two distinct groups. Previous studies have illustrated prognostic differences within the highly aggressive UM tumors, classifying monosomy 3 UM tumors into two groups [18]. Recently, loss of *BAP1* has been shown to be associated with DNA methylation repatterning needed for UM to gain malignant potential [62]. Although there is no functional proof that *FOXD1* and *BAP1* interact, due to the distinct *FOXD1* expression pattern in UM it might be involved in diverting from a low-risk to high-risk tumor. In line with previous published gene expression profiles, we illustrate that the identified transcription regulators follow a similar gene expression profile as seen in the GEP-test (Figure 6). To prove that *FOXD1* is either a driver or bystander in high-risk UM, functional assays must be performed to illustrate its mechanism. However, in this article we provide proof that *FOXD1* can reliably be used as a biomarker in UM. *FOXD1* expression has been assessed in multiple cancer types and is often correlated to a poor prognosis [63,64,65,66]. In vitro and in vivo experiments elucidated *FOXD1* as an oncogene involved in proliferation, dedifferentiation, migration, and radio- and chemoresistance [67,68,69,70,71]. For instance, in oral squamous cell carcinoma, *FOXD1* promotes the epithelial–mesenchymal transition (EMT) by activation of *SNAI2* [72]. Functional work on *FOXD1* in cutaneous melanoma elucidated its capability to induce the expression of the tumor-specific isoform Rac Family Small GTPase 1B (*RAC1B*) to enhance melanoma migration and invasion [38]. Additionally, upon *BRAF* inhibitor treatment, *FOXD1* was shown to directly bind the CTGF promoter to promote dedifferentiation and therapy resistance [64]. In line with published data on EMT, we assessed *FOXD1* expression with known EMT inducers and repressors described in squamous cell carcinoma and cutaneous melanoma. However, unlike in squamous cell carcinoma and cutaneous melanoma, these EMT-like patterns are not clearly or significantly correlated with *FOXD1* in UM (Appendix A). It seems that EMT-like patterns of UM are different than what is currently been described in literature concerning the downstream effects of *FOXD1*. For instance, a loss of *CDH1* has been describe to be necessary for EMT [73], while in the GEP test, *CDH1* is used as an upregulated biomarker for class 2 UM.

## 5. Conclusions

In this study we found the expression of *FOXD1* to be almost exclusively in *BAP1*-mutated UM. *BAP1* mutations are associated with early metastatic disease [74,75], although its underlying mechanism remains unclear. Our analysis shows that most *BAP1*-mutated UMs with *FOXD1* expression are correlated with poor survival. This correlation was also seen using immunohistochemistry, although the correlation was not as strong as on the RNA level. To understand what role *FOXD1* plays in UM, it is necessary to perform functional assays. Due to the striking correlation of *FOXD1* expression in BAP1-mutated UM, functional assays elucidating its function and targets could provide therapeutic targets for high-risk UM.

## Figures and Tables

**Figure 1 cancers-14-03668-f001:**
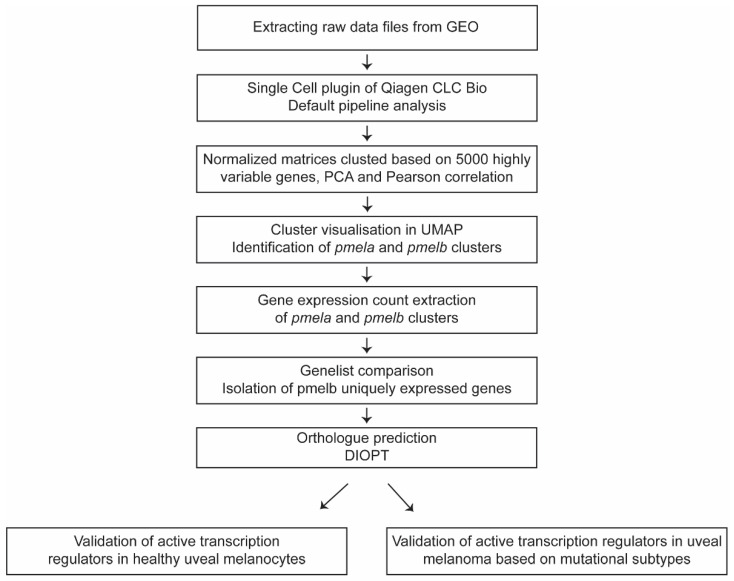
Schematic overview of the single-cell workflow from raw data files to validation of gene expression in uveal melanoma.

**Figure 2 cancers-14-03668-f002:**
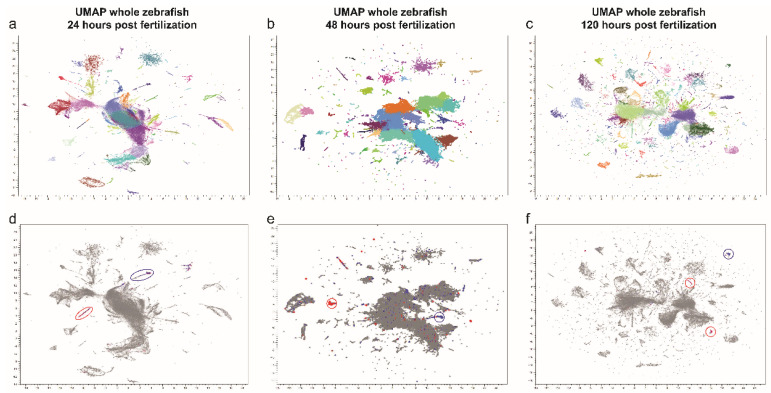
Uniform manifold approximation and projection (UMAP) plot depicting single-cell transcriptomes from whole zebrafish larvae aged (**a**) 24, (**b**) 48 and (**c**) 120 h post fertilization (hpf). Each dot represents a single cell. Melanocyte cluster gene expression was isolated from *pmelb*-negative cells (blue) and *pmelb*-positive cells (red) and combined for further analysis (**d**–**f**).

**Figure 3 cancers-14-03668-f003:**
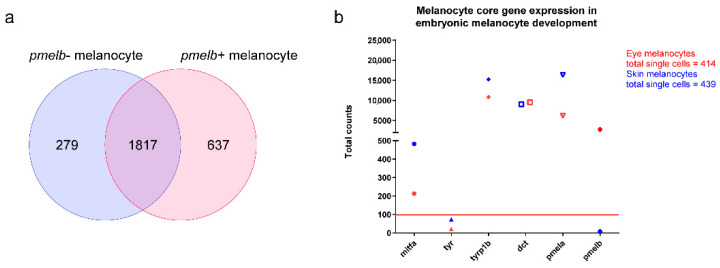
(**a**) Venn diagram of the melanocytic clusters in the zebrafish atlas. Clusters were divided into *pmelb*-positive and *pmelb*-negative melanocytes. A total of 1817 genes were expressed in both clusters, whereas 637 genes were uniquely expressed in the *pmelb*-positive cluster and 279 genes in the *pmelb*-negative cluster. (**b**) Total counts of melanocyte markers in the *pmelb*-positive and *pmelb*-negative clusters. An overall high total count of most melanocyte markers was assessed in both clusters, while *pmelb* is only expressed in one cluster.

**Figure 4 cancers-14-03668-f004:**
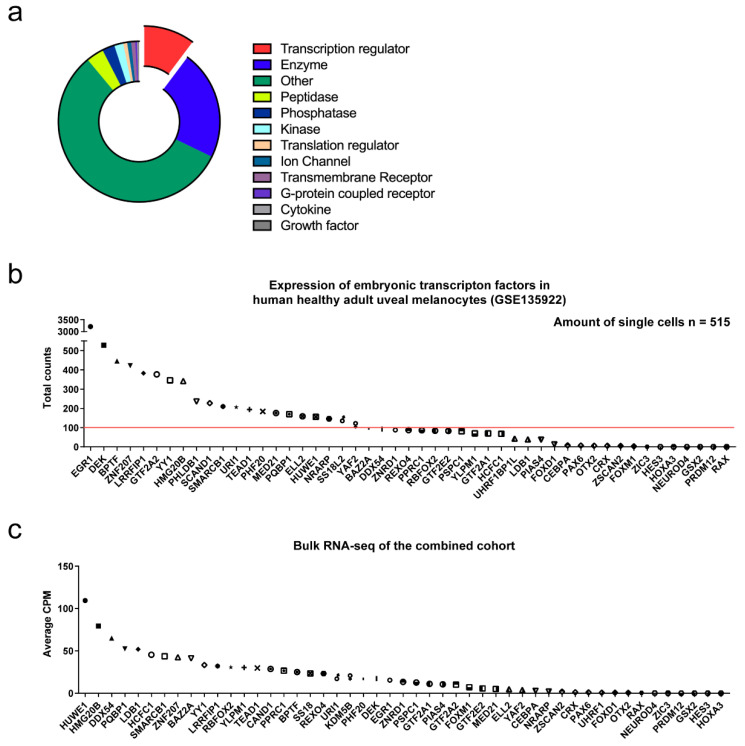
(**a**) Gene function annotation from IPA, with including 49 transcription regulars. Expression of the 49 transcription regulator genes (expressed in early-developed zebrafish larvae) in (**b**) human healthy uveal melanocytes and (**c**) UM. Twenty-one genes were expressed in human healthy uveal melanocyte whereas the majority of the genes are expressed in UM. Red line = gene expression threshold of 100.

**Figure 5 cancers-14-03668-f005:**
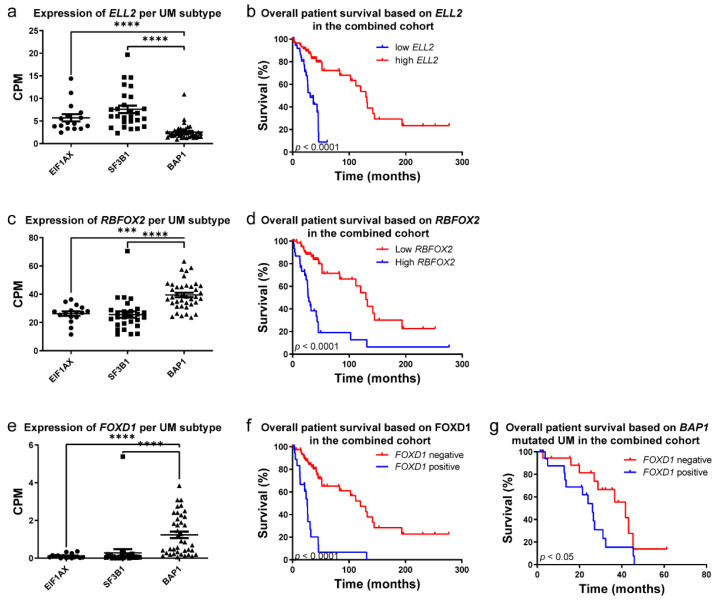
(**a**) Expression of *ELL2* in UM, grouped per mutational subtype and (**b**) survival plot of expression in the combined (ROMS and TCGA) cohorts. (**c**) Expression of *RBFOX2* in UM, grouped per mutational subtype and (**d**) survival plot of expression in the combined (ROMS and TCGA) cohorts. (**e**) Expression of *FOXD1* in UM, grouped per mutational subtype and (**f**) survival plot of expression in the combined (ROMS and TCGA) cohorts and (**g**) *BAP1*-mutated UM within the combined cohort. ***, *p* < 0.001; ****, *p* < 0.0001.

**Figure 6 cancers-14-03668-f006:**
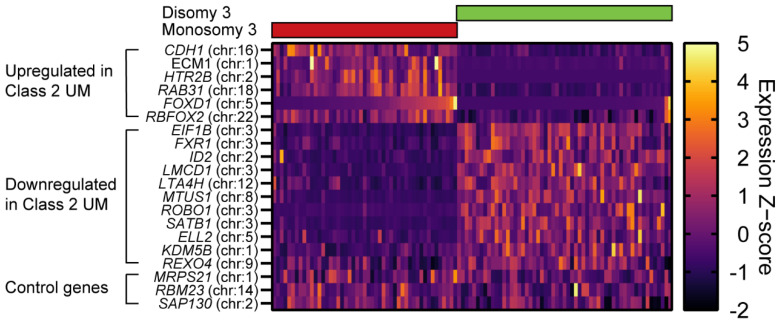
Heatmap using Z−scores of the GEP test of UM, including *FOXD1*, *RBFOX2*, *ELL2*, *KDM5B* and *REXO4* on UM samples. Samples were sorted based on disomy 3 or monosomy 3 status. *FOXD1* and *RBFOX2* clustered similarly to upregulated genes, whereas *ELL2*, *KDM5B* and *REXO4* were more similarly clustered to downregulated genes.

**Figure 7 cancers-14-03668-f007:**
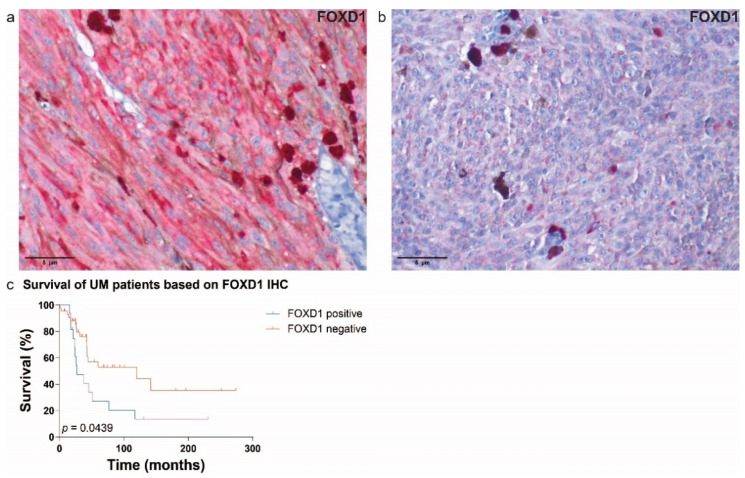
Immunohistochemical expression of FOXD1; (**a**) FOXD1-positive UM and (**b**) FOXD1-negative UM (scale 5 µm). (**c**) The survival plot of FOXD1 expression on IHC in the combined cohorts (discovery and replication cohorts). The expression of FOXD1 was correlated with a poorer survival (n = 59, *p* = 0.0439).

## Data Availability

The data from The Cancer Genome Atlas (TCGA) and repository NCBI GEO Datasets are publicly accessible. Our ethics committee does not allow sharing of individual patient or control genotype information in the public domain.

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
