# Peer review of "FOXD1 Is a Transcription Factor Important for Uveal Melanocyte Development and Associated with High-Risk Uveal Melanoma"

_cancers, 2022, doi:10.3390/cancers14153668_

Round 1
Reviewer 1 Report
The manuscript “FOXD1 is a transcription factor important for uveal melanocyte 2 development and associated with high risk uveal melanoma” addresses potential altered transcription factor expression as a pro-metastatic factor in uveal melanoma. The authors exploit the Zebrafish model, normal human melanocyte and human uveal melanoma. They identify several candidate transcription factors and analyze in some detail FOXD1.
This study is preliminary and no firm conclusions can be drawn. FOXD1 expression in the developing human uvea is unknown. The Zebrafish model hints at a role in embryogenesis but developmental expression is one of the most divergent features in the evolution of vertebrates.
The correlation of the expression of FOXD1 with metastatic risk could not be confirmed at the protein level and its function in the formation of metastasis is therefore not documented since mere mRNA expression levels are unlikely to have functional consequences. The authors suggest that larger sample collection could reveal such a correlation and they should do this analysis.
Most events are linked to BAP1 mutational status. The paper does not allow to distinguish whether the differential transcription factor expression is a driver or bystander effect.
The expression of these genes in correlation to monosomy of chromosome 3, another important prognostic factor, is not reported.
Accession numbers of the cohorts should be indicated.
Author Response
Dear reviewer,
Thank you for your time and input towards this manuscript.
Please see the attachment.

Reviewer 2 Report
Dear authors,in figure 5 you have to correct didascalies.
Author Response

(The authors gave the same response as above.)

Reviewer 3 Report
Overall, the idea of the work is interesting. The findings of the work could have merit in the related field. Just, few concerns should be addressed.
Introduction
- Line 51: “UM has an incidence of up to 8 per million in Europe”, the authors should update this figure based on a recent citation as it was Since 2007 (15 years ago).
Materials and Methods
- It will be hard to follow the authors in their elaborations in this section, in particular for non-specialty related readers. It is highly recommended to simplify this great work by providing an informative detailed workflow that illustrates the RNA-seq work stages with its sequential applied datasets and control measures.
- Figure S1 was not mentioned in the text, and it was not informative. Please replace by a main Figure as recommended above.
- Line 151: what was the dilution factor of the applied antibody?
- What was the type of controls the authors applied in their IHC analysis?
Results
- In general the figure quality is very low, in particular Figures 1 and 5.
- Lines 190/191: please revise the sentence: “A total of 1817 genes were expressed in both 190 genes,”
- Line 263: “a) FOXD1 negative UM” please revise to match panel “a”
Minor comments
- line 63: “all UM all arise...” needs revision
- Line 255: it is better to avoid starting the statements with figures.
Author Response

(The authors gave the same response as above.)

Round 2
Reviewer 1 Report
The authors have adequately replied to almost all issues raised. I still believe that it should not be a major problem to retrieve sufficient samples to definitely either confirm or reject the hypothesis that FOXD1 protein expression is linked to UM metastatic risk.
I think this important since the whole study only shows correlations not functionally explained causation. We would at least like to kknow whether the correlation also holds on the protein level in a significant manner.
